# Spring Plates as a Valid Additional Fixation in Comminuted Posterior Wall Acetabular Fractures: A Retrospective Multicenter Study

**DOI:** 10.3390/jcm12020576

**Published:** 2023-01-11

**Authors:** Domenico De Mauro, Giuseppe Rovere, Lorenzo Are, Amarildo Smakaj, Alessandro Aprato, Umberto Mezzadri, Federico Bove, Alessandro Casiraghi, Silvia Marino, Gianluca Ciolli, Simone Cerciello, Giuseppe Maccagnano, Giovanni Noia, Alessandro Massè, Giulio Maccauro, Francesco Liuzza

**Affiliations:** 1Orthopaedics and Traumatology Unit, Fondazione Policlinico Universitario Agostino Gemelli, IRCCS, 00168 Rome, Italy; 2Department of Orthopaedics and Traumatology, Università Cattolica del Sacro Cuore, 00168 Rome, Italy; 3Orthopaedics and Traumatology Unit, Azienda Ospedaliero Universitaria Città della Salute e della Scienza, 10126 Turin, Italy; 4Department of Orthopaedics and Traumatology, Università degli Studi di Torino, 10125 Turin, Italy; 5Orthopaedics and Traumatology Unit, ASST Grande Ospedale Metropolitano Niguarda, 20162 Milan, Italy; 6Orthopaedics and Traumatology Unit, ASST degli Spedali Civili, 25123 Brescia, Italy; 7Orthopaedics Unit, Department of Clinical and Experimental Medicine, Faculty of Medicine and Surgery, University of Foggia, Policlinico Riuniti di Foggia, 71122 Foggia, Italy

**Keywords:** acetabular fracture, posterior wall, ORIF, spring plate, multicenter study

## Abstract

Background: The posterior wall fracture is the most frequent pattern of acetabular fractures. Many techniques of fixation have been described in the literature and involve plates, screws, or a combination of both. This study aims to investigate the clinical and radiological outcomes of spring plates in the treatment of comminuted posterior wall acetabular fractures. (2) Methods: A retrospective multicenter (four level I trauma centers) observational study was performed. Patients with a comminuted posterior wall acetabular fracture treated with a spring plate (DePuy Synthes, West Chester, PA) were included. Diagnosis was made according to the Judet and Letournel classification. Diagnosis was confirmed with plain radiographs in an antero-posterior view and Judet views, iliac and obturator oblique views, and thin-slice CT with multiplanar reconstructions. (3) Results: Forty-six patients (34 males and 12 females) with a mean age of 51.7 years (range 19–73) were included. The most common mechanism of injury was motor vehicle accident (34 cases). In all cases, spring plates were placed under an overlapping reconstruction plate. The mean follow-up was 33.4 months (range 24–48). The mean period without weight-bearing was 4.9 weeks (range 4–7), and full weight-bearing was allowed at an average of 8.2 weeks (range 7–11) after surgery. (4) Conclusions: According to the present data, spring plates can be considered a viable additional fixation of the posterior wall acetabular fractures.

## 1. Introduction

A posterior wall fracture is the most frequent pattern of acetabular fractures [1,2]. According to the Judet and Letournel classification system, they represent approximately 25% of all acetabular fractures, and dislocation of the femoral head occurs in more than 85% of these patients [3,4].

Several fixation techniques are reported in the literature including plates, screws, or combinations of both. In the case of comminution, the posterior wall fragments may be very thin and small, and they can hardly be fixed with an overlying reconstruction plate or screws. Moreover, fractures close to the acetabular rim are not easily fixed with a plate alone and the attempt of a lag screw fixation can increase the risk of joint penetration [5,6,7].

In such cases, spring plates have been advocated. Mast et al. [8], in 1989, were the first to propose the concept of spring plates for adjunctive fixation of comminuted posterior wall fractures. At the beginning, a three hole one-third tubular plate was used. It was modified by pre-bending to achieve a slightly convex shape and by cutting the central portion of its distal hole to leave two adjacent prongs. Both these prongs were finally bent at 90 degrees toward the plate undersurface of the plate to compress the small periarticular fragments creating a buttress effect. Since then, a pre-contoured device whose distal end has two hooks engaging the non-articular cortical surface of the fracture fragment has been developed. These plates are placed only after the reduction in any posterior wall fragments and can be used whether under an overlapping reconstruction plate or in addition to a reconstruction plate and interfragmentary screws, orthogonally to the articular fragments. The buttressing effect of adding spring plates to the recon plate on the small periarticular fragments makes the anatomical reduction more feasible than the recon plate alone [9,10,11,12,13].

This study aims to investigate the clinical and radiological outcomes of spring plates in the treatment of comminuted posterior wall acetabular fractures. According to our experience, outcomes of patients treated through spring plates are expected to be good, either clinically or radiologically.

## 2. Materials and Methods

A retrospective multicenter observational study was performed, in four I level trauma centers in Italy. The study was performed according to the principles of the Declaration of Helsinki. Due to the purely retrospective and observational design of the study, local Ethics Committees confirmed that no ethical approval was required. Informed consent was obtained from all subjects involved in the study. For the study aims, all the posterior wall acetabular fractures between February 2018 and February 2020 who underwent internal fixation with the use of spring plate (DePuy Synthes, West Chester, PA, USA) were included. Patients with incomplete data, follow-up shorter than 24 months, pathological fractures of the acetabulum, associated femoral head fractures [14], history of homolateral acetabular fractures were excluded. Diagnosis was made according to the Judet and Letournel classification, on plain radiographs in antero-posterior view and Judet views, iliac and obturator oblique views, and thin-slice CT with multiplanar reconstructions (Figure 1). 

Patients were sampled using data available in the hospital database. The data collected were divided into preoperative (gender, age, mechanism of injury, time interval between the trauma and the surgery, radiological exams), intra-operative (duration of the surgery and implant type and size), and postoperative (complications, delay in weight-bearing, clinical outcomes at follow-up).

All the patients in the four centers were operated by senior surgeons (FL, FB, UM, AA, AC), all active members of the Italian Society for the Traumatology of the Pelvis (A.I.P.). All patients were set in prone position on a radiolucent carbon table to allow intra-operative radiological visualization without interference. Kocher-Langenbeck approach, which is the accepted gold standard in the treatment of posterior wall fractures [15], was performed in all cases, The surgery was carried out using SPS Matta Pelvic System (Stryker Trauma AG, Selzach, Switzerland), and according to the fracture pattern, different implants were used. Spring plates were the implants considered in the study. In all cases, spring plates were placed under an overlapping reconstruction plate. The prongs should not be placed too close to the articular edge or into the labrum because this would damage the articular surface. The medial hole of the spring plate is fixed with one or two 3.5 mm cortical screws. As the screws are tightened, the slightly convex shape allows the plate to contour the underlying bone and the prongs to stabilize the joint fragments. After surgery, no weight-bearing was indicated for at least 4 weeks.

Postoperative clinical and radiographic examinations were carried out at 1, 3, 6, 12 months, and then 24 months after the surgery. The quality of surgical reduction was assessed in AP and Judet views of X-rays by measuring the residual postoperative displacement and according to the radiographic criteria by Matta [5], they were classified as anatomical (0–1 mm of displacement), imperfect (2–3 mm), or poor (>3 mm). Clinical outcomes were evaluated with the Merle d’Aubigne and Postel (MAP) [16] and modified Harris Hip Score (mHHS) [17] to assess function and satisfaction at 24 months follow-up. Modified HHS is a clinical score evaluating the hip function, with the following values: <70 poor result, 70–79 fair result, 80–89 good result, and >90 excellent result. MAP score has a range from <7 (poor) to 11–12 (excellent). Weight-bearing was also evaluated. 

Data were analyzed for descriptive statistics as mean or median for continuous variables and frequency distribution (%) for categorical variables. Fisher’s exact test and the qui-square test were used to study categorical variables.

## 3. Results

Forty-six patients (34 males and 12 females) with a mean age of 51.7 years (range 19–73) were available at a mean follow-up of 33.4 months (range 24–48). The most common mechanism of injury was motor vehicle accident (34 cases), and among them 13 were pedestrians hit by a vehicle (Table 1). The fracture was judged to be anatomical in 36 cases, imperfect in 7 cases, and poor in 3 cases. At the last radiographic FU, no delayed unions or malunions were observed, as well as screw penetration into the joint (Figure 2). In one case, implant loosening was observed due to the patient’s poor compliance with the rehabilitation protocol. The patient underwent total hip replacement (THR). 

One patient developed femoral head necrosis 12 months after surgery; in this case, THR was performed as well.

The mean period without weight-bearing was 4.9 weeks (range 4–7) and full weight-bearing was reached after an average of 8.2 weeks (range 7–11) after surgery, with complete Range of Motion (ROM) of the affected hip. Post-traumatic osteoarthritis (OA) was evidenced during the radiological follow-up in seven patients (15.2%), affecting the clinical outcome with pain and joint stiffness. These results were largely validated by all the multidimensional observational scales applied in this study, considering pain and function: the mean MAP was 10.2 (range 5–12) and the mean mHHS was 84.9 (range 59–94). Finally, no cases of neurological injuries, wound infections, or heterotopic ossifications were recorded (Table 2).

## 4. Discussion

This study aims to increase the awareness about the use of spring plates in the treatment of posterior wall acetabular fractures, as a valid additional option to better perform an anatomical and stable reduction. Posterior wall acetabular fractures, isolated or associated with other acetabular lesions, represent a challenge for trauma surgeons [16]. Anatomic reduction and stable fixation are crucial to restore joint congruity, avoid poor functional results, and reduce the risk of late cartilage degeneration [17]. Posterior wall fractures are often associated with comminution and marginal impaction, so that stable fixation of the small periarticular fragments with screws and plates is difficult and the risk of joint penetration or additional surgery-related comminution is high [18]. 

Matta et al. [5] underlined the strong correlation between the quality of the surgical reduction and the clinical outcome and assessed that even a satisfactory reduction, with a residual postoperative displacement of more >2 mm, leads to altered load distribution in the hip and progressive post-traumatic arthritis. In these situations, the use of a spring plate was recommended to ensure fixation of the small fragments and to avoid intra-articular penetration of screws that could be necessary to secure such fragments close to the articular surface [9]. 

In the present study, anatomic reduction with a postoperative displacement < 1 mm was achieved in 36 out of 46 cases (76.1%), whereas in 8 cases (17.4%) the reduction was imperfect (2–3 mm) and in just three cases (6.5%) was poor (>3 mm). 

At the last clinical follow-up, most patients showed a good functional recovery with no or mild pain and no or minimal hip stiffness. As evidenced in our study, the values reached at the MAP score and at the mHHS were satisfactory, and mostly positive according to the clinical outcome and the pain control, and moreover, it did not affect quality of life. 

However, a significant number of patients developed post-traumatic OA. As a matter of fact, comminuted fractures of the posterior wall are at high risk of post-traumatic OA, even with a good reduction after the surgery. This risk is even higher when marginal impaction is present. The positive impact of spring plates in terms of construct stability must be emphasized. Richter et al. investigated the biomechanics of spring plate fixation in association with standard plates and demonstrated good primary stability and solid fixation in the treatment of comminuted posterior wall fragments thanks to its dynamic buttress effect [9]. Goulet et al., indeed, found that the addition of spring plates increased the load to failure when compared to a reconstruction with a standard plate alone [6].

Pease et al. performed prolonged cyclic loading protocols on models and recorded the displacement of the fragments: the reconstruction rim plate, alone or with lag screws, was the most mechanically reliable construct [19]. Nevertheless, it was correlated to a higher risk of screws’ joint penetration or other complications. In addition, fracture displacement was greater in the spring plate model compared to the reconstruction plate with lag screws through it, but no significant difference in overall construct stiffness was observed between models. Furthermore, spring plates provided a safe and precise fixation with a reduced risk of screws’ joint penetration, as these were located far away from the joint.

The present study has some notable limitations. Firstly, the retrospective design and the lack of a control group may affect the conclusions of the study, and also do not allow a clear statement regarding the use of spring plates as routinary additional fixations in comminuted posterior wall acetabular fractures. A case series study design was obliged since a control group is hard to set given the strict and limited inclusion criteria (comminuted posterior walls fractures) and the ethical concerns about the non-use of spring plates in patients who can take advantage from its use. 

Even the retrospective study design was made necessary due to the rare occurrence of acetabular fractures compared to other fractures. However, a prospective study can help build stronger evidence in the literature.

Secondly the study population is quite limited; however, this is the consequence of the uncommon occurrence of comminuted posterior wall fractures. The only available study with a larger number of patients is the one by Lee et al., which included 52 patients [10]. However, in that study, it was not really a spring plate that was implanted, rather a three hole one-third tubular plate modified by the surgeon. Additionally, the available literature concerning spring plates is relatively poor, and the majority of data are focused on self-made spring plates or other off-label implants. Ziran et al. applied a modified distal radial T-plate as a spring plate [20]. It should allow for a wider contact area close to the articular rim by cutting the distal portion of the T-plate to create multiple prongs. 

Recently, Cho et al. proposed an alternative fragment-specific fixation technique using multiple 2.7 mm variable-angle locking compression plates (VA-LCPs), designed for distal radius, particularly indicated for highly comminuted posterior wall fractures, or involving the superior dome of the acetabulum. The angular stable fixation of each fragment was achieved with locking screw fixation through the VA locking plate holes. The technique allows a more stable fixation of small fracture fragments compared to spring plates, without the necessity of an overlapping reconstruction plate. However, screw joint penetration may potentially happen in peripheral fractures and the biomechanical analysis of such a construct is lacking [21]. 

Although these limitations of the present study have several relevant aspects, the study population is homogeneous, with the same surgical indication for all patients, the spring plate was a pre-contoured ready-to-implant device in all cases, eliminating any bias regarding the surgeon’s manipulation of the implants, and radiological evaluation was available in all cases at FU to confirm the extent of the reduction and the healing of the fractures. Multicenter collaboration allows a consistent number of cases to be collected, showing one of the largest populations in the papers regarding this topic. Moreover, the endorsement of the A.I.P. and the cooperation among highly experienced and highly skilled pelvic trauma surgeons, provides the study with not only a solid consistency and uniformity in the correct diagnosis and classification, but also in treatment choices. 

## 5. Conclusions

According to the present study and the analysis of the literature, it is confirmed that spring plates are an effective adjunctive fixation in the treatment of comminuted posterior wall acetabular fractures, where a lag screw through the rim plate cannot be applied. Their use is associated with a reduced risk of joint penetration and implant failure or the excessive soft tissue dissection necessary to use additional pelvic reconstruction plates. This is crucial to achieve a reduction as much as anatomically possible and, consequently, to ensure good long-term clinical outcomes.

## Figures and Tables

**Figure 1 jcm-12-00576-f001:**
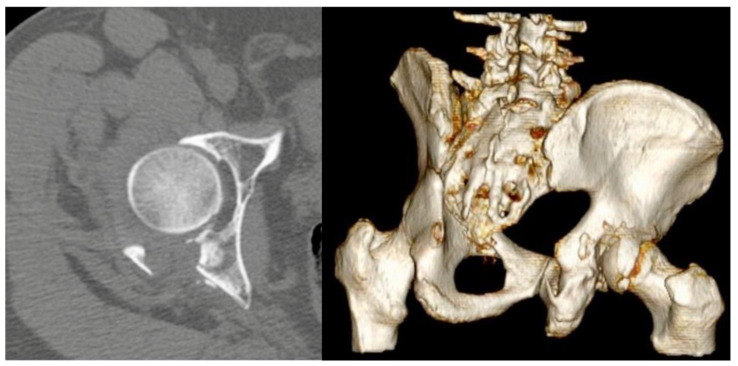
Computed tomography (CT) axial scans and 3D reconstruction showing comminuted posterior wall fracture with marginal impaction.

**Figure 2 jcm-12-00576-f002:**
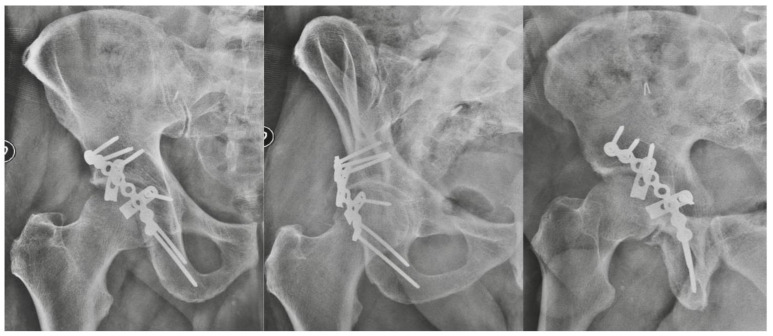
X-rays on AP view and Judet views showing spring plates used in addition to a reconstruction plate and interfragmentary screws.

**Table 1 jcm-12-00576-t001:** Demographic data of the patients from the participating centers.

Participating centers	4	
n. of patients	46	
Age	51.7	
Sex	Male	34 (73.9%)
	Female	12 (26.1%)
Traumatic Mechanism	Road Accident	34 (73.9%)
	Falls from heights	12 (26.1%)
Diagnosis	Posterior Wall	46 (100%)

Diagnosis according to Judet and Letournel classification.

**Table 2 jcm-12-00576-t002:** Clinical and radiological data after follow-up.

Mean follow-up	33.4 months		
Radiological outcome	Anatomical	78.3%	36 patients
	Imperfect	15.2%	7 patients
	Poor	6.5%	3 patients
Average full weight-bearing	8.2 weeks after surgery		
Average Merle d’Aubigne score	10.2 ± 1.7		
Average modified Harris Hip Score	84.9 ± 6.5		

Radiological outcome according to Matta’s criteria.

## Data Availability

The data presented in this study are available on request from the corresponding author. The data are not publicly available due to privacy.

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
