# Peer review of "Spring Plates as a Valid Additional Fixation in Comminuted Posterior Wall Acetabular Fractures: A Retrospective Multicenter Study"

_jcm, 2023, doi:10.3390/jcm12020576_

Round 1

Reviewer 1 Report

I congratulate the authors for conducting a retrospective multicenter study including four level I trauma centers in Italy to present outcome of spring plates in posterior wall acetabular fractures. 46 patients with minimum follow-up of 24 months (mean 33 months) could be included in the case series. I will outline my comments in a point-by-point basis below

Abstract

·      Well written, easy to follow, contains the main points

Introduction

·      Well written introduction

·      If possible you can state a hypothesis following the aim (line 68)

Material and Methods

·      Line 71: Despite not needing ethical approval, did you obtain patient consent to use their data?

·      Line 89-90: How many surgeons did the cases? 

·      Line 97: Who is the manufacturer of the reconstruction plate? Matta or Synthes? 

·      Line 105: Do you have a reference for the Matta criteria? I guess it is ref number 5.

·      Line 107: At what timepoint have you evaluated MAP and mHHS? You should tell the reader here or in the results section.

Results

·      Line 120: Do you have a reference for the Matta criteria? I think it is ref nr 5. Besides,I think the sentence from lines 119-121 can be deleted because the same information was given in lines 105-106.

·      Line 138: “After surgery, no weight-bearing was indicated for at least 4 weeks.” The aftercare should be described in material and methods 

·      Lines 140-144: This outcome description is very vague. The data should be available from your mHHS. Can you give mean +- SD values for those or present the data in a separate table? 

·      Lines 148-150: Could you obtain the MAP and mHHS for the complete cohort? (N= 46)?

Discussion

·      Very well written discussion providing good overview of the current literature, easy to follow

·      Line 154-155: I am very glad to see excitement for the multicenter collaboration and to present a large cohort in the literature. However, I would recommend moving this sentence down where you described the limitations. The multicenter aspect might be named as strength. Further, I guess, the sentence should be rewritten because it was not the aim of the study to demonstrate the strengths of a multicenter collaboration. 

·      Line 156-159: The same goes for this sentence, should be moved to the limitations section and named as strength. 

·      Line 159-161: I am afraid you can’t state this because you had no control group as you state perfectly in Lines 203-204. It would rather apply to weaken the sentence a bit. However, it is correct that the study could show satisfactory results using a Spring Plate but we don’t know if it is advantageous compared to reconstruction plate only because of the missing control group. 

·      Lines 154-161: You should start the discussion differently either be reiterating the study question, weakness in the current literature and purpose or by naming your strongest result (= good clinical and radiographic outcome, low complication rate) or both combined. 

·      Line 185: Can you correlate the rate of OA with the amount mal-reduction or is there no correlation? This would be very interesting for the reader and probably the citation rate of the article as one would expect malreduction to be predisposing factor but it is unknown what the exact threshold is. At what mean timepoint did OA occur?

·      Line 208-209: I think you should modify this sentence (“the ethical concerns about the non-use of Spring Plates in patients who can take advantage from its use.“) because, as you stated later, there is not enough evidence for Spring Plates yet. Consecutively, you can’t say a comparative study would not be justified ethically. 

·      Line 234-235: You must clarify because in lines 102-105 you wrote that you used x-ray to analyze postop reduction. At what timepoint did you use CT?

o   See “The quality of surgical reduction was assessed in AP and Judet views X-rays by measuring the residual postoperative displacement and according to the radiographic criteria by Matta” (Lines 102-105)

Conclusion

·      Line 237: Did you mean analysis “of” literature?

·      Line 241: I would modify this sentence because you wrote in your materials and methods that you used reconstruction plates as well. Subsequently, there would be any advantage in terms of soft tissue stripping. 

Reviewer 2 Report

This is an interesting undertaking and study. While there are some limitations, namely the small sample size, the findings are of interest given the limited data regarding this topic. However, there are some modifications that may improve the manuscript:

Results:

Most of the patients showed a good to excellent functional 141 recovery with no pain or mild pain not affecting their quality of life. – Please clarify this statement and include statistics and data where possible.

As for ROM of the 142 affected hip, most patients reported no or slight hip stiffness during normal activities: in 143 most cases, there was a little decrease especially in the internal rotation, not bringing to 144 functional impairment. – Please clarify this statement and include statistics and data where possible.

Discussion:

Line 217: Please replace “Anyway” with a more appropriate term, such as “Additionally”. Please also correct the spacing in the beginning of that sentence.  

Furthermore, without the inclusion of specific scores etc is it difficult to assess the significance of these results and conclusions.  While there is no comparison group for p values, percentages are important to support the findings of this study. Please include where possible. 

Round 2

Reviewer 2 Report

Thank you for taking the time to address the inquiries. Although the study has some interesting findings, the addition of objective data and clear statistics would improve this study. The authors have made claims of "majority" and have mentioned pain vs. no pain, but these claims were not backed up with data. Are there any specific data that can be added? 

If not, the authors should explain this in the limitations section as to why this information cannot be produced or unavailable. 
